# CPPF, A Novel Microtubule Targeting Anticancer Agent, Inhibits the Growth of a Wide Variety of Cancers

**DOI:** 10.3390/ijms21134800

**Published:** 2020-07-07

**Authors:** Ho Jin Han, Chanmi Park, Joonsung Hwang, Thimmegowda N.R., Sun-Ok Kim, Junyeol Han, Minsik Woo, Shwetha B, In-Ja Ryoo, Kyung Ho Lee, Hyunjoo Cha-Molstad, Yong Tae Kwon, Bo Yeon Kim, Nak-Kyun Soung

**Affiliations:** 1Anticancer Agent Research Center, Korea Research Institute of Bioscience and Biotechnology, Ochang, Cheongju 28116, Korea; hjhan@kribb.re.kr (H.J.H.); springcm@naver.com (C.P.); hwangj1@kribb.re.kr (J.H.); nrthimmegowda@gmail.com (T.N.R.); sunok@kribb.re.kr (S.-O.K.); jyhan@kribb.re.kr (J.H.); wms9460@kribb.re.kr (M.W.); shwethatg16@gmail.com (S.B.); ijryoo@kribb.re.kr (I.-J.R.); leekh@kribb.re.kr (K.H.L.); hcha@kribb.re.kr (H.C.-M.); 2Department of Biomolecular Science, University of Science and Technology, Daejeon 34113, Korea; 3College of Pharmacy and Medical Research Center, Chungbuk National University, Osong, Cheongju 28160, Korea; 4Protein Metabolism Medical Research Center and Department of Biomedical Sciences, College of Medicine, Seoul National University, Seoul 03080, Korea

**Keywords:** microtubule target agent, anticancer drug, skin cancer prevention/animal model, multidrug resistance, CPPF

## Abstract

In the past, several microtubule targeting agents (MTAs) have been developed into successful anticancer drugs. However, the usage of these drugs has been limited by the acquisition of drug resistance in many cancers. Therefore, there is a constant demand for the development of new therapeutic drugs. Here we report the discovery of 5-5 (3-cchlorophenyl)-N-(3-pyridinyl)-2-furamide (CPPF), a novel microtubule targeting anticancer agent. Using both 2D and 3D culture systems, we showed that CPPF was able to suppress the proliferation of diverse cancer cell lines. In addition, CPPF was able to inhibit the growth of multidrug-resistant cell lines that are resistant to other MTAs, such as paclitaxel and colchicine. Our results showed that CPPF inhibited growth by depolymerizing microtubules leading to mitotic arrest and apoptosis. We also confirmed CPPF anticancer effects in vivo using both a mouse xenograft and a two-step skin cancer mouse model. Using established zebrafish models, we showed that CPPF has low toxicity in vivo. Overall, our study proves that CPPF has the potential to become a successful anticancer chemotherapeutic drug.

## 1. Introduction

Chemotherapy has been a keystone of cancer treatment for many years. Over time, numerous therapeutic drugs have been developed [1]. Two successful chemotherapeutic drugs are vinblastine and paclitaxel. Vinblastine was isolated in 1958 and has been used to treat many cancers, including lymphoma, testicular cancer, breast cancer and choriocarcinoma [2] Paclitaxel was first isolated in 1971 and was approved for cancer treatment by the FDA in 1992. It has been broadly used to treat solid tumors such as breast cancer and metastatic non-small cell lung cancer [3]. Both vinblastine and paclitaxel are microtubule targeting agents (MTAs) [4].

Microtubules are a major component of the cytoskeleton. These proteins consist of a heterodimer of α and β tubulin and are constantly polymerizing and depolymerizing to form microtubule filaments. These dynamic microtubules exert critical functions in many cellular processes, such as cellular component trafficking, cell shape maintenance and mitotic spindle formation [5,6,7]. Since cancer cells divide rapidly, they undergo more mitosis and chromosomal duplication than normal cells. Therefore, microtubule functions are more crucial, making tubulin a desirable target for cancer treatment [8].

MTAs are classified into six groups depending on their tubulin-binding site. These groups are taxanes, lauramide, colchicine, vinca alkaloids, maytansine and pironetin. Of these, only pironetin binds α-tubulin while the others bind β-tubulin at distinct sites [9,10]. Interestingly, colchicine binds β-tubulin at the interface of the α and β-tubulin dimer [9,11]. MTAs are also divided into two groups based on their effect on microtubule formation. They are either microtubule stabilizer agents (MSAs) or microtubule destabilizer agents (MDAs). Taxane and lauramide are MSAs, whereas colchicine, vinca alkaloids, maytansine and pironetin are MDAs [9,12]. However, MSAs and MDAs have opposite effects on microtubules, both inhibit tubulin dynamics resulting in mitotic arrest and eventually apoptosis [13]. The FDA has approved many MTAs for clinical usage; however, their use has been limited by the development of drug resistance in many cancers. Therefore, the identification of novel MTAs that can be used to treat these resistant cancers is a key priority [14,15].

According to a recent report, skin cancer is one of the most common forms of cancer in the United States [16] The two-step skin cancer model is a well-known carcinogen-induced spontaneous tumor model used to study skin cancer. In this model, cancer is initiated by a single topical treatment of 7,12-dimethylbenz[α] anthracene (DMBA), and promotion is induced by repeated treatments of phorbol ester 12-O-tetradecanoylphorbol-13-acetate (TPA) [17]. Using this two-step model, the tumor prevention effect of a compound can be studied. The compound in question is combined with the TPA treatment, and the number of tumor occurring events are measured.

Three dimensional (3D) cultured spheroids are an emerging research tool in the field of cancer. These cultures are superior to standard cell cultures because they model the interplay between a tumor and its microenvironment. This is particularly important because the microenvironment of a tumor greatly influences tumor growth and survival [18].

In this study, we screened small molecule library from the Korea Chemicals Banks looking for potential anticancer drugs using cell based antiproliferation assay and we identified of a novel microtubule targeting anticancer agent, 5-(3-chlorophenyl)-N-(3-pyridinyl)-2-furamide (CPPF). We confirmed its anticancer effect on a variety of cancer cell lines using both two dimensional (2D) and 3D cell culture systems. In addition, we show that CPPF is an effective anticancer agent against known multidrug-resistant (MDR) cell lines. Furthermore, mouse xenograft and two-step skin cancer mouse models reveal that CPPF has both cancer inhibition and cancer prevention effects in vivo. Therefore, our data suggest that CPPF has the potential to be a new cancer therapeutic drug.

## 2. Results

### 2.1. CPPF Inhibits Cell Proliferation of Various Cancer Cell Lines

We screened a small-molecule compound library to discover novel antiproliferative agents and identified CPPF (Figure 1A). We first tested the antiproliferative ability of CPPF on HeLa cells, a widely used cervical cancer cell line (Figure 1B). CPPF-induced growth inhibition at a concentration of 1 µM after 24 h. Over the next 4 days, CPPF significantly inhibited cell growth in a concentration-dependent manner (Figure 1C) with an IC_50_ of 700 nM (Figure 1B). Next, we tested the growth inhibition effect of CPPF on various cancer cell lines derived from cervical, breast, leukemia, hepatoma, stomach, lung, prostate and skin cancers. Each cell line was treated with various concentrations of CPPF (0.1–10 µM) for 4 days. Using MTT cell viability assays, we measured the IC_50_ value of CPPF for each cell line. The IC_50_ values ranged from 0.2 to 4 µM. CPPF induced an antiproliferative effect on all cancer cell lines tested with high potency in HeLa, JurkeT, Whel3, Snu484, U937 and NCI-H1299 cells (Appendix A).

### 2.2. CPPF Inhibits Growth of 3D Tumor Spheroid Cultures

To examine the anticancer effect of the CPPF in a culture system that better mimics in vivo conditions, we generated HeLa cell spheroid using a 3D cell culture system. These spheroids were treated with various concentrations of CPPF (0.25–2 µM), and images were captured daily to calculate the tumor spheroid area (Figure 2A). After 3 days, growth inhibition was observed at the lowest CPPF concentration Figure 2B,C). At concentrations above 0.5 µM, cultures showed the growth of many single cells separate from the spheroids. As the concentration of CPPF increased, this phenotype became more prominent (Figure 2A). In summary, CPPF was able to inhibit growth in both cancer cell lines and 3D culture systems.

### 2.3. CPPF Causes Mitotic Arrest

In response to CPPF treatment, most HeLa cells acquired a round shape similar to that seen during mitosis. Therefore, we hypothesized that CPPF could affect the mitotic process [19]. We first used FACS analysis to examine the effect of CPPF on the cell cycle. CPPF at a concentration of 1 µM or higher had a significant increase in the number of G2/M phase cells (Figure 3A,B). To determine whether CPPF treatment leads to mitotic arrest, we examined various mitotic marker proteins such as PLK1, Cyclin B1, and CDC25C using western blot analysis. At a concentration of 1 µM or higher, CPPF treatment led to an accumulation of these protein markers indicating mitotic arrest (Figure 3C) [20]. Since several tubulin target compounds are known to cause mitotic arrest by inducing abnormal spindle formation, we examined spindle formation in cells treated with CPPF using immunofluorescence. Cells treated with 1-µM CPPF showed the formation of multipolar spindles rather than the expected bipolar spindles (Figure 3D), suggesting that CPPF affects tubulin formation [21].

### 2.4. CPPF Targets Microtubules and Inhibits Tubulin Dynamics

After observing that CPPF-induced multipolar spindle formation, we hypothesized that CPPF may target microtubules by binding tubulin. Using molecular docking models and known MTAs, we determine that CPPF could overlap the tubulin-binding site for colchicine (Figure 4A) [22]. To determine whether CPPF exhibits effects similar to colchicine, a microtubule depolymerizing agent, microtubule polymerization/depolymerization assays were performed. Microtubules naturally polymerized at 37 °C. Some MTAs such as paclitaxel accelerates tubulin polymerization, whereas other such as vinblastine induces tubulin depolymerization. As expected, control samples showed natural polymerization, and paclitaxel-treated samples rapidly polymerized and became saturated. However, the CPPF-treated sample had a slow polymerization rate and did not saturate, similar to vinblastine (Figure 4B).

Microtubule networks are essential for cellular trafficking and cell cycle function. When microtubule polymerization/depolymerization are disrupted, tubulin dynamics are reduced, causing multiple cellular problems [23]. To test the influence of CPPF on tubulin dynamics, we carried out microtubule regrowth assays [24]. HeLa cells were incubated on ice for 30 min as a cold shock to disturb microtubule networks. Then, warm media with or without CPPF was added. To visualize microtubule formation under CPPF treatment, cells were fixed and stained to observe α-tubulin and γ-tubulin. Microtubule regrowth (represented by α-tubulin, green signal) starts from the centrosome (represented by γ-tubulin, red signal) in the cytosol. Immediately after cold shock, γ-tubulin was detected in both the control and CPPF samples, but α-tubulin was seen only in the control. In the control, microtubule bundles formed at 1 min, and after 3 min, the microtubule bundles returned to a normal state. However, microtubule regrowth was suppressed in the sample treated with 1 µM CPPF at the 1 min time point and significantly decreased at later time points compared to the control (Figure 4C). These result show that CPPF targets microtubules, causing depolymerization thus disrupting microtubule networks.

### 2.5. CPPF Induces Apoptosis

MTAs induce prolonged mitotic arrest and other cellular stresses that eventually result in cell death by apoptosis [25]. To determine if CPPF induces apoptosis, we used Annexin V and PI double staining FACS analysis. Cells were treated with CPPF, stained for Annexin V and PI and then analyzed by FACS to measure the percent of apoptotic cells. Both Annexin V positive or Annexin V and PI double-positive cells represent apoptosis. The number of apoptotic cells significantly increased in samples treated with 1 µM CPPF (Figure 5A). To verify that CPPF causes apoptosis, we utilized western blot analysis to observe apoptosis marker proteins PAPP, caspase 3, caspase 7, caspase 8 and caspase 9, whose cleaved forms represent apoptosis. The results show that CPPF treatment led to the accumulation of cleaved marker proteins in a dose-dependent manner (Figure 5B). Therefore, CPPF treatment leads to apoptosis.

### 2.6. CPPF Is Effective Against Multidrug Resistance Cell Lines

However, MTAs have been successfully used as chemotherapy agents, one major factor that limits their effectiveness is the development of drug resistance. To test the effectiveness of CPPF against MDR cancers, we utilized MDR cancer cell lines MCF7/ADR and K562/ADR [26,27]. Paclitaxel and colchicine MTAs showed a 153.6-fold and 30.9-fold increase in their IC_50_, respectively, in K562/ADR cells compared to the nonresistant parent cell line K562. In addition, paclitaxel and colchicine showed a 393.2-fold and 37.1-fold increase in their IC_50_, respectively, in MCF7/ADR cells compared to MCF7 cells. When CPPF was tested against these cell lines, the results showed that CPPF had a similar IC_50_ in both K562 and K562/ADR cell lines and a 4-fold decrease in its IC_50_ in MCF7/ADR cells compared to MCF7 cells (Table 1). Therefore, CPPF is effective against cells that have acquired resistance to other well-known MTAs.

MDR cell lines were treated with 0 to 10 µM CPPF for 4 days. Then, cell viability was measured and calculated in an MTT assay. Data presents IC_50_ (µM) by Mean ± Standard deviation of three independent experiments. The resistant factor (in parenthesis) was calculated by the ratio of the IC50 of the resistant to that of the parental cells.

### 2.7. CPPF Has Low Toxicity As Shown By Zebrafish Testing

In developing a newly synthesized drug, toxicity is always a major concern. To test the toxicity of CPPF, we utilized a zebrafish model. After treating 5-day-post-fertilization (DPF) zebrafish with various concentrations of CPPF (1–20 µM) for 72 h, no zebrafish death and no morphologic changes were detected at concentrations up to 10 µM, which is about 15 times greater than the IC_50_ of CPPF in Hela cells (Figure 6A).

In addition to lethality, one common side effect seen in drug treatment is neurotoxicity. To test the potential neurotoxicity of CPPF, we conducted two different animal neurotoxic behavior experiments. First, a zebrafish locomotion test was done to detect any changes in movement [28]. Individual 5 DPF zebrafish were transferred to wells of a 96-square-well plate and tracking movement. At concentrations of 5 µM or lower, CPPF treatment did not significantly change fish movement. However, a reduction in fish movement occurred at a CPPF concentration of 10 µM (Figure 6B).

The second behavior test was a zebrafish color preference test. Zebrafish naturally favor the color blue over yellow [29]. The test utilized a blue–yellow chamber to evaluate the effect of CPPF on zebrafish color preference. CPPF was tested at three concentrations (2.5, 5, 10 µM). No significant change in color preference was seen even at the highest CPPF concentration (10 µM) (Figure 6C). In conclusion, CPPF treatment at a concentration of 10 µM did not affect fish survival or color preference and had only a slight impact on fish movement.

### 2.8. CPPF Is Effective at Treating and Preventing Cancer in Mice Models

Next, we tested the ability of CPPF to exert its anticancer properties in vivo. To do this, an in vivo mouse xenograft model was used. The results show that CPPF suppressed tumor size by approximately 20% compared to the vehicle control (Appendix A). In addition to treatment, cancer prevention is also a critical issue. To test if CPPF affected cancer prevention, we used a carcinogen-induced spontaneous skin cancer model, which is a two-step skin cancer model. In essence, this model uses two stages of chemical application to the skin for the initiation and promotion of cutaneous tumors [30]. To induce the skin tumor, a single application of the initiator mutagen, DMBA, is used. To promote tumor growth, repeated applications of the proinflammatory phorbol ester (TPA) are done. In this experiment, we treated mice with TPA only or TPA combined with CPPF. After 3 weeks, the average number of tumors was decreased by almost 50% in the CPPF versus control group. Therefore, CPPF demonstrates both cancer treatment and prevention effects in two-step skin cancer mice models (Figure 7A,B).

## 3. Discussion

During the development of novel anticancer drugs, many factors need to be considered, such as efficacy, toxicity and potential drug resistance [31,32,33]. In this study, we introduce a small molecule called CPPF that was selected from a small-molecule compound library for its ability to inhibit cancer growth. In the process of determining its viability as an anticancer drug, we first examined its efficacy in vitro. Using 2D culture models and MTT assays, we showed that CPPF was effective in inhibiting the growth of a variety of cancer cell lines from diverse origins. The characteristic of these CPPFs is that if developed as an anticancer drug, the limitations of therapeutic application to cancer due to origin specificity may not be such a big problem. In addition to this study, most anticancer drug studies use a 2D culture model. However, this method is an insufficient indicator of drug efficacy in vivo. Since they cannot accurately reflect the microenvironment of a tumor, such as oxygen and nutrient gradients, which are factors that affect tumor growth and survival. Therefore, we utilized a 3D tumor spheroid culture model to further examine CPPF drug efficacy [18,34]. The results confirmed CPPF’s ability to inhibit cancer growth in not only the 2D culture system but also the 3D’s. Therefore, this result showed that CPPF is suitable for development as an anticancer agent. In the course of our in vitro efficacy experiments, we noticed that cells treated with CPPF developed morphology similar to that of mitotic cells. Therefore, we hypothesized that CPPF may cause mitotic arrest. Using FACS, western blot and immunofluorescence analysis, we showed that CPPF causes mitotic arrest through the abnormal formation of multipolar spindles. This led us to focus on CPPF’s interaction with tubulin. Using molecular docking models to predict potential binding sites for CPPF on tubulin, we discovered that CPPF has the potential to bind tubulin at a site shared by the well-known MTAs, colchicine. Colchicine is known to cause the depolymerization of tubulin. To see if CPPF also had this phenotype, we conducted a tubulin polymerization assay to measure CPPF influence on microtubule polymerization. Our results clearly showed that, like colchicine, CPPF prevented or delayed tubulin polymerization, which is how CPPF induces mitotic arrest. Prolonged mitotic arrest and inhibition of tubulin dynamics result in cellular stress accumulation that eventually causes apoptotic cell death [35]. Therefore, we examined whether CPPF treatment results in apoptosis. Using FACS and western blot analysis to look at apoptotic markers, the results conclusively showed that CPPF causes apoptosis in cancer cells. In summary, CPPF is a new MTA whose mode of action is through the induction of mitotic arrest and eventual apoptosis of cancer cells. Many MTAs are well-known chemotherapy agents. However, their use has been limited due to the rise of drug resistance in cancer patients. Drug resistance may be caused by a wide variety of factors such as altered cell cycle checkpoint proteins, increased efflux, decreased uptake, DNA damage repair, cell death inhibition, drug target alteration and drug inactivation [36]. Since CPPF shares a binding site on tubulin with the MTA colchicine, it may be possible that cells already resistant to colchicine would be resistant to CPPF. Therefore, we tested CPPF effectiveness against MDR cell lines MCF7/ADR and K562/ADR, which are resistant to both paclitaxel and colchicine MTAs. The results showed that CPPF shares a similar phenotype with most the tubulin targeting compounds, but it can overcome MDR that limits other MTAs. However, we were not able to point out how the compound could bypass the mechanism of drug resistance. However, due to the possibility that small molecules can bind various target proteins, we could expect to be able to have a tubulin-independent mechanism responsible for overcoming MDR, such as inhibition of MDR-related proteins or drug efflux proteins. In summary of drug efficacy, our compound has an anticancer effect in various cancer cells including MDR. However, it has anticancer effect on higher concentration than other clinically used MTAs. Therefore, we will make derivative compound to optimize drug delivery and improve system stability in order to meet clinical standards. The toxicity of a compounds is a critical issue in drug development. If CPPF were toxic at effective concentrations, then its use as an anticancer drug would be negated. Therefore, we tested CPPF toxicity using commonly accepted zebrafish models [37] which have several experimental benefits such as a high fertility rate, transparent in early development phase and fast adult development compared to other vertebrate models [38]. In this animal system, even though the fish showed a slight change in movement analysis, in lethality, morphology or color preference, we observed no serious change at 10 µM, approximately 15 times higher than the IC_50_ of CPPF in Hela cells. In addition, we could not notice any significant problems in CPPF treated mice by injection or topical treatment.

Therefore, the use of CPPF as an anticancer drug would not be prohibited due to its toxicity. To test the anticancer effects of CPPF in vivo, we utilized two in vivo mouse models. We used a two-step skin cancer model to investigate the cancer prevention effects of CPPF. Skin cancer is one of the major cancers [39,40]. The number of skin cancer patients is predicted to accelerate around the world due to the depletion of the ozone layer [41]. Since the development of skin cancer is known to be preventable [42], we tested the potential preventive effects of CPPF. Our skin cancer model clearly shows that CPPF has a significant cancer prevention ability. Therefore, CPPF has the potential to be used as a drug to reduce the number of skin cancer patients in the future. We also used a xenograft model to further confirm CPPF’s anticancer effect in vivo. The result shows that CPPF can induce growth inhibition of tumors in vivo. With improvements in drug delivery and system stability, we expect even stronger anticancer effects from CPPF treatment in the future.

## 4. Materials and Methods

### 4.1. Synthesis of CPPF

Synthesis of 5-(3-chlorophenyl)-N-(pyridin-3-yl)furan-2-carboxamide hydrochloride (CPPF) was outlined in reaction scheme 1 (Included in Appendix A). The synthesis of the title compound CPPF was started with commercially available 3-chlorophenylboronicacid (1) and 5-bromo-2-furoic acid (2) starting materials by Suzuki coupling reaction using cesium carbonate (Cs_2_CO_3_) as base and palladium-tetrakis(triphenylphosphine) (Pd(PPh_3_)_4_) catalyst to obtain the intermediate compound 5-(3-chlorophenyl)furan-2-carboxylic acid (3) in 55% yield. The title compound 5-(3-chlorophenyl)-N-(pyridin-3-yl)furan-2-carboxamide was readily synthesized by the coupling of 5-(3-chlorophenyl)furan-2-carboxylic acid (3) with 3-aminopyridine (4) using EDCI coupling agent and finally converted to its corresponding hydrochloride salt 5-(3-chlorophenyl)-N-(pyridin-3-yl)furan-2-carboxamide hydrochloride in 70% yield by using 2-M hydrogen chloride solution in diethyl ether. The structure of newly synthesized molecule CPPF was determined by ^1^H NMR, ^13^C NMR and ESIMS analyses.

Please see the online Appendix A and Methods for reaction scheme and detailed experimental procedure and ESIMS, ^1^H NMR, ^13^C NMR spectra.

### 4.2. Chemicals and Antibodies

Paclitaxel was purchased from EMD Millipore (Billerica, MA, USA). Colchicine, nocodazole and vinblastine were purchased from Sigma-Aldrich (St. Louis, MO, USA). Antibodies against Plk1 (# sc-17,783), Cdc25C (# sc-5620), GAPDH (# sc-25877), and PARP-1 (# sc-7150) were purchased from Santa Cruz (Dallas, TX, USA). Cyclin B1 (# 4138), phospho-Histone H3 (ser10) (# 9706), caspase-8 (# AM46) and caspase-9 (# 9502) antibodies were purchased from Cell Signaling Technology (Denvers, MA, USA). Caspase-3 (# IMG-144A) antibody was purchased from IMGENEX (San Diego, CA, USA). caspase-7(# AAM-137) was purchased from Stressgen (San Diego, CA, USA). Anti α-tubulin antibody (# T6074), anti γ–tubulin (# T5192) antibody and Hoechst 33342 were purchased from Sigma-Aldrich (St. Louis, MO, USA). Moreover, Texas red and Alexa 488 conjugated secondary antibodies were purchased form Molecular Probes (Thermo Fisher Scientific, Waltham, MA, USA). For animal experiment, acetone, DMBA and TPA were purchased from Sigma-Aldrich.

### 4.3. Cell Lines

HeLa, MCF7, Hep3B, HepG2, A431, K562, Jurkat, U937, P388, EL4, WehI3, A549, NCI-H1299 and PC3 cells were purchased from American Type Culture Collection (ATCC, Manassas, VA, USA). SNU484 and SNU601 cells were obtained from the Korean Cell Line Bank (KCLB, Seoul, Korea). MCF7/ADR and K562/ADR cells were produced by the Bio Evaluation center of KRIBB [43]. All medium contain 100 U penicillin/(100 µL/mL) streptomycin from Corning Cellgro (Corning, NY, USA) and 10% fetal bovine serum from Invitrogen (Carlsbad, CA, USA). Cells were incubated at 37 °C in a humidified incubator with 5% CO_2_. DMEM and RPMI 1640 media were purchased from HyClone (Logan, UT, USA).

### 4.4. MTT Assay

Cells were seeded in 96-well plates at a concentration of 1.2–4.5 × 10^3^ cells/well in 100 µl. After 17 h, varying concentrations (0, 0.1 µM, 0.3 µM, 1 µM, 5 µM,10 µM) of CPPF were used to treat cells. Each concentration was done in triplicate. Cells were incubated for 4 days and then, cell viability was determined by adding 10 µl of cytox (LPS solution, Daejon, Korea) to each well followed by incubation at 37 °C for 2 h. Absorbance was measured at 450 nm using a SpectraMax 190 microplate reader (Molecular Devices, San. Jose, CA. USA). IC_50_ was calculated in a nonlinear regression analysis using Graphpad Prism 6.0 program (San Diego, CA, USA).

### 4.5. Three-Dimensional Culture

To make a spheroid formation, HeLa cells were seeded in 96-well round-bottom low attachment plates (Shimadzu Sci. Kyoto, Japan) at a concentration of 1.0 × 10^4^ cells/well in 50 µl and incubated at 37 °C in 5% CO_2_ for 24 h. Then, varying concentrations of CPPF in 50 µl of media were added to each well. Every day, cell images were captured using a Zeiss Primovert microscope (Carl Zeiss, Oberkochen, Germany). Spheroid areas were calculated using Image J (National Institutes of Health, Bethesda, MD, USA) [34]

### 4.6. Flow Cytometric Analysis

For cell cycle phase analysis, HeLa cells were seeded in 12-well plates at a concentration of 2.0 × 10^4^ cells/well and incubated at 37 °C for 17 h. Then, cells were treated with varying concentrations of test compounds and incubated for an additional 17 h. Next, cells were harvested and washed with PBS. Then, cells were stained with propidium iodide (PI) according to the Cycle Test Plus DNA Reagent kit protocol (BD Biosciences. San Jose, CA, USA). Cells were analyzed using a FACS Calibur instrument (BD Biosciences).

To analyze apoptosis, cells were seeded in 6-well plates and incubated at 37 °C for 17 h. Cells were then treated with varying concentrations of test compounds and incubated for an additional 48 h. The APC Annexin V apoptosis detection kit with PI (Thermo Fisher Scientific) was used to detect apoptosis. The staining was done according to the manufacturer’s protocol. Flow cytometric analysis was performed using CytoFLEX (Beckman Coulter, Brea, CA, USA) and measured using FlowJo software (BD Biosciences).

### 4.7. Western Blot

Cells were lysed using RIPA buffer (150-mM NaCl, 50-mM Tris-HCl pH 8.0, 1% NP-40, 0.5% sodium deoxycholate and 0.1% SDS) containing 1-mM DTT, 1-mM Na_3_VO_4_, protease inhibitors and phosphatase inhibitors (Sigma-Aldrich). Cell lysates were centrifuged and the supernatant was transferred to new tubes. The appropriative amounts of cell lysate were subjected to SDS-PAGE and then transferred to PVDF membranes (Bio-Rad. Hercules, CA, USA). Membranes were incubated in blocking buffer (Bio-Rad) and specific antibodies solutions. All membranes were detected using an ECL kit (Thermo Fisher Scientific) and exposed to film.

### 4.8. Tubulin Depolymerization Assay

Tubulin polymerization assays were performed according to the manufacturer’s protocol (Cytoskeleton, Denver, CO, USA). Briefly, 200 µl of pure tubulin protein was resuspended in 420 µl of tubulin polymerization buffer (80-mM PIPES (pH 6.9), 2-mM MgCl_2_,0.5-mM EDTA, 3.75% glycerol and 1-mM GTP) to a final concentration of 5 mg/mL. This was done on ice. Then, 100 µl of the reaction mixture was added to the wells of a 96-well plate containing 10 µl of CPPF, paclitaxel, vinblastine or DMSO (control) solutions. The final compound concentrations for CPPF were 5-µM and 10 µM. All other compounds had a final concentration of 10 µM. The samples were mixed, and tubulin assembly was measured by continuous monitoring of the turbidity change at 340 nm at 37 °C [24,44].

### 4.9. Computer Modeling Study

To examine the binding mode of CPPF with respect to the impairment of tubulin activity, we conducted docking simulations at the active site. Three-dimensional atomic coordinates were extracted from the X-ray crystal structure of tubulin (PDB code: 1SA0) to act as the receptor model [11] Gasteiger–Marsili atomic charges were determined for all the protein and ligand atoms to calculate the electrostatic interactions between tubulin and CPPF. Docking simulations to address the binding mode of CPPF were then carried out with a modified version of the AutoDock program (Scripps Research Institute, La Jolla, CA, USA) whose performance has been well-appreciated for various target proteins [45].

### 4.10. Immunofluorescence Microscopy

HeLa cells were seeded in 24-well plates containing coverslips at a concentration of 1.0 × 10^4^ cells/well and incubated at 37 °C for 17 h. Then, cells were treated with varying concentrations of test compounds and incubated for an additional 17 h. Cells were then fixed using 4% paraformaldehyde. Coverslips were permeated with cold methanol and were blocked with BSA. Anti α-tubulin and anti γ-tubulin antibodies were incubated for 2 h, Texas red and Alexa 488 conjugated antibodies were incubated for 1 h. Coverslips were washed with PBS and incubated in Hoechst 33342 (0.5 µg/mL in PBS) for 10 min. Then, immobilized Fluoro-Gel (Electron Microscopy Sciences, Hatfield, PA, USA) mounting solution was added to the coverslips; and images were obtained on a fluorescence microscope.

### 4.11. Cold Shock Tubulin Growth Assay

HeLa cells were seeded on glass coverslips and incubated at 37 °C for 18 h. Cells were then treated with 1-µM CPPF and placed on ice for 30 min to depolymerize microtubules. Next, cells were transferred to prewarmed (37 °C) media for 1, 3 or 5 min; permeabilized with 4% paraformaldehyde, and then fixed with methanol. Tubulin bundles were observed using anti α-tubulin and anti γ-tubulin immunostaining [24].

### 4.12. Zebrafish Toxicity Test

Male and female zebrafish were housed and mated under standard conditions (28 °C water temperature, feeding twice every day). After eggs were fertilized, they were transferred to 100-mm dishes containing embryo medium. All experiments were conducted on eggs 5 DPF. Five fish from 5 DPF larvae were grouped in 12 wells and treated with 0 to 20 µM of CPPF. After 24, 48 or 72 h, images were captured, and live fish counted. Experiments were done in triplicate.

### 4.13. Zebrafish Behavioral Test

Individual 5 DPF eggs were transferred into wells of a 96 square well plate containing 300 µl of embryo medium and left to adapt for 1 h at 28 °C. Then 0 to 10-µM CPPF was added to a final medium volume of 400 µl (*n* = 6). Larva behaviors were tracked using the DanioVision video tracking system (Noldus, Wageningen, Netherlands) and total distances moved (mm) were measured and analyzed using the Etho vision XT software (Noldus).

In the color preference test, larvae were placed in a two-color maze chamber (yellow/blue). Ten of each larva per group were treated in 0 to 10 µM CPPF in 3 mL of embryo medium after incubation for 1 h in the dark. After adaptation, the larvae were exposed to 1 h of 123 lux light and recorded. Every 10 min, fish in the blue area were counted and analyzed.

### 4.14. Carcinogen-Induced Skin Cancer

Chemical carcinogenesis was induced in skin as previously described [46,47]. The dorsal skin area of the 6-week-old FVB/N male mice was shaved 2 days before the start of the experiment. each group consisted of 20 mice. The initiation of tumorigenesis was performed by a single topical treatment with 100 µg of DMBA in 0.2 mL of acetone over 1 week and then, tumor promotion was extended by treatment with 5 µg of TPA in 0.2 mL of acetone, twice weekly for 15 weeks. CPPF at a concentration of 500 nM in 0.2 mL of acetone was applied topically 30 min after TPA treatments. Skin tumors such as papillomas appearing on the dorsal skin were recorded every week during the experimental period, with only those having a diameter of 2 mm considered as positive. At the end of 15 weeks, the mice were euthanized with CO_2_ and sacrificed. All animal studies were conducted in accordance with the guidelines approved by the Institutional Animal Care and Use Committee of KRIBB (permit number: KRIBB-AEC-16081).

### 4.15. Statistical Analysis

Statistical evaluation of the cell growth assay (Figure 1C), 3D culture Area calculation (Figure 2C), FACS analysis for cell cycle (Figure 3B), FACS analysis for apoptosis (Figure 5A), Zebrafish toxicity and movement (Figure 6B,C) and skin cancer number (Figure 7B) was carried out with a nonparametric *t*-test using Graphpad Prism 6.0 program. Moreover, all graphs were created using GraphPad Prism 6. Data are given as mean ± SD and considered significant if *p* ≤ 0.05 (* *p*-value < 0.05, ** *p*-value < 0.005, *** *p*-value < 0.001, **** *p*-value < 0.0001). NS represent Not significant.

## 5. Conclusions

In summary, CPPF effectively inhibits the growth of a variety of cancer cell lines, can overcome MDR limiting other MTAs currently in clinical use, is effective in vivo and has low toxicity. However, it has less of an apoptotic effect and higher IC_50_ than other clinically used MTAs. Therefore, we will make derivative compounds not only to increase the apoptotic effect and lower the IC50 but also to optimize drug delivery and improve system stability in order to meet clinical standard.

## 6. Patents

KR10-1721490 B1.

## Figures and Tables

**Figure 1 ijms-21-04800-f001:**
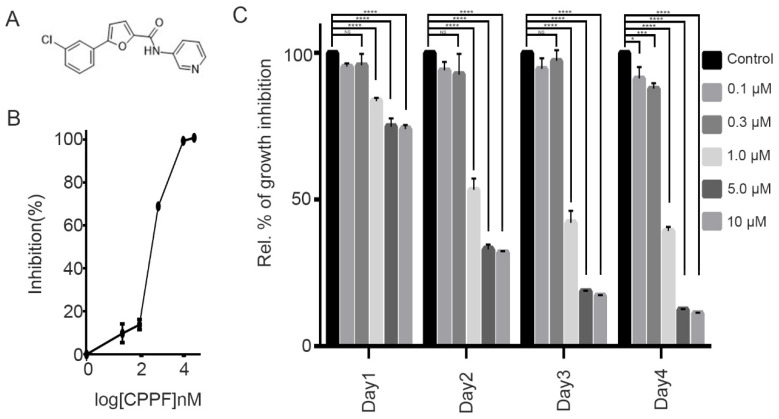
Structure and effect of 5-5 (3-cchlorophenyl)-N-(3-pyridinyl)-2-furamide (CPPF) on the cell proliferation of various cancer cell lines. (**A**) Chemical structure of CPPF; (**B**) HeLa cells were seeded in a 96-well plate. Cells were treated with CPPF for 4 days. Cell growth was measured at 450 nm absorbance using an MTT assay. Data were analyzed using GraphPad Prism software. Error bars represent means ± SD from three independent experiments (**C**) HeLa cells were seeded in a 96-well plate and treated with different concentrations of CPPF. Cell growth was measured at 1 day, 2 days, 3 days and 4 days after treatment using an MTT assay. Error bars represent means ± SD from three independent experiments, * *p*-value < 0.05, *** *p*-value < 0.001 and **** *p*-value < 0.0001, NS: Not significant.

**Figure 2 ijms-21-04800-f002:**
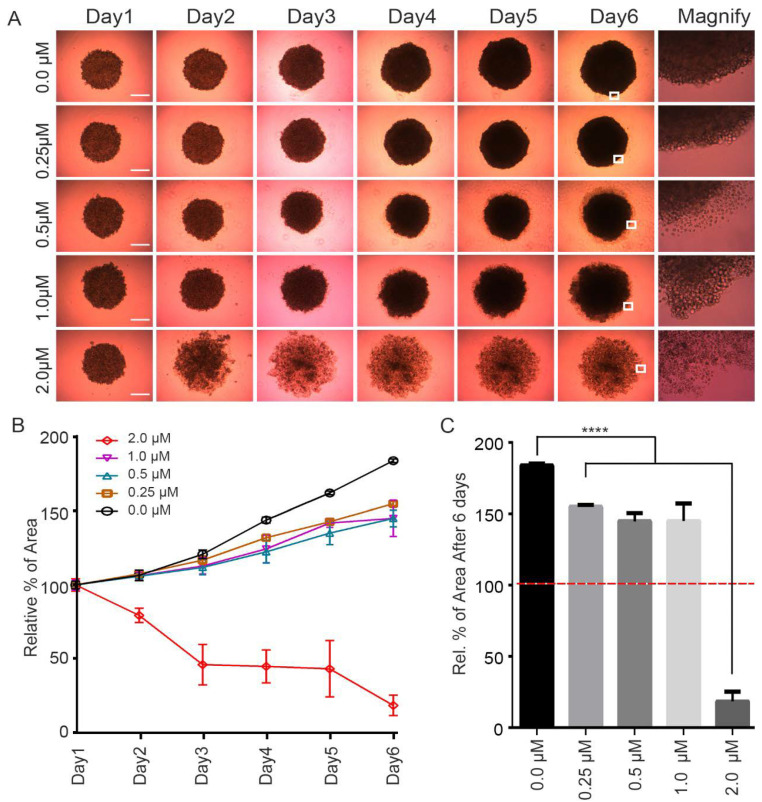
CPPF anticancer effect on 3D culture system. (**A**) HeLa cells were seeded in 96-well round-bottom low attachment plate and treated with different concentrations of CPPF. Images were captured using a Zeiss Primovert microscope. The scale bar is 100 µm; (**B**) Spheroid areas from D were calculated using Image J. Data presented as the mean ±SD of three independent experiments; (**C**) The spheroid area on day 6. The red line represent start point of spheroid area for CPPF treatment. Error bars represent means ± SD from three independent experiments, **** *p*-value < 0.0001.

**Figure 3 ijms-21-04800-f003:**
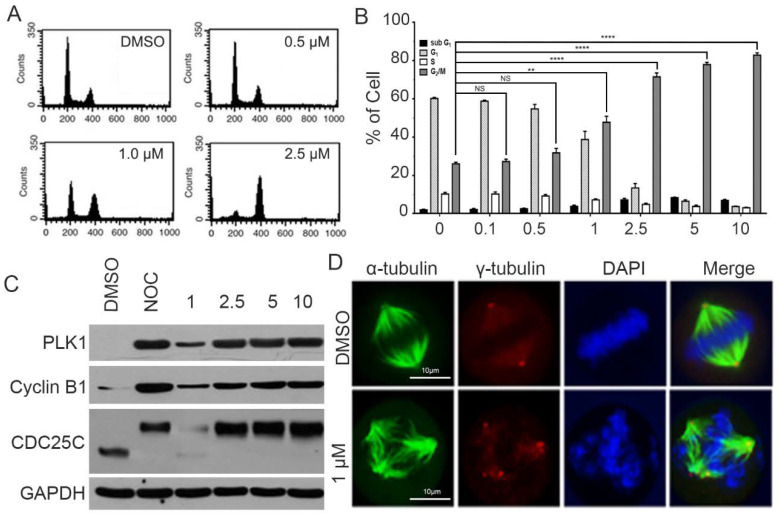
CPPF-induced cell cycle arrest during mitosis in HeLa cells. (**A**) HeLa cells were treated with various concentrations of CPPF for 17 h. Cells were harvested and stained with propidium iodide (PI). Cell cycle distribution was analyzed by flow cytometry; (**B**) Plots from A were analyzed to represent the percentage of cells in each phase graphically. Error bars represent means ± SD from three independent experiments, ** *p*-value < 0.005 and **** *p*-value < 0.0001. NS: Not significant; (**C**) HeLa cells were treated with DMSO (control) or various concentrations of CPPF for 17 h. Western blot data showed the accumulation of mitotic arrest marker proteins such as PLK1, cyclin B1 and cdc25c. These data are representative of three independent experiments. Nocodazole (NOC) treatment was at a concentration of 100 ng/mL; (**D**) Relative percentages of mitotic cells after CPPF treatment. Seventeen hours after treating HeLa cells with CPPF, tubulin was visualized by α-tubulin (green) staining and the centrosome was visualized by γ- tubulin (red) staining. DNA was stained using Hoechst 33342 (blue). Images were captured by a Zeiss microscope.

**Figure 4 ijms-21-04800-f004:**
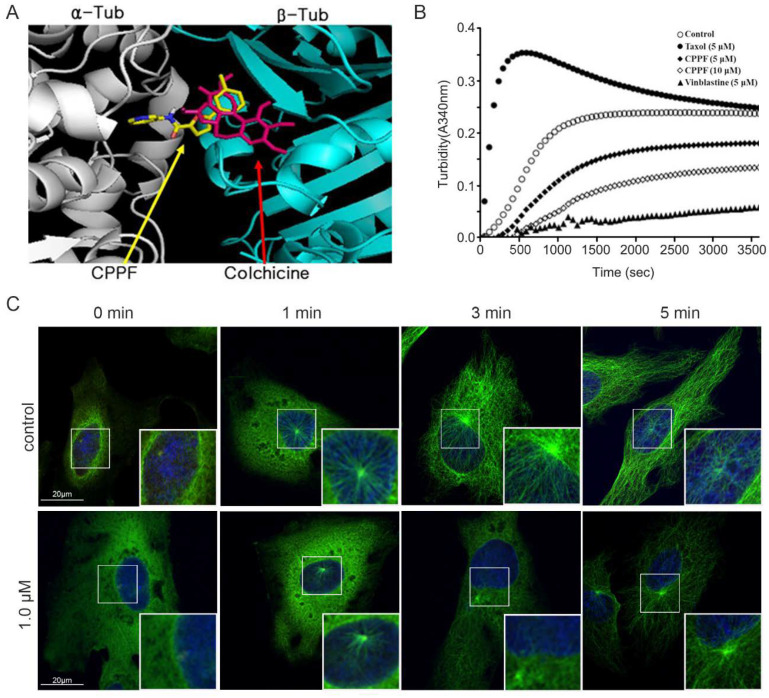
CPPF targets microtubules. (**A**) Computer modeling of the CPPF binding site on tubulin; (**B**) Bovine tubulin was incubated at 37 °C with or without compounds and time-dependently measured at 340 nM to track tubulin polymerization; (**C**) HeLa cells were treated with DMSO control or 1 µM CPPF. Then cells were incubated in ice-cold media for 30 min. After, the media was replaced with warm media and cells were time-dependently fixed using paraformaldehyde. Cells were stained with α-tubulin (green), γ-tubulin (red) and DNA (blue). Images were captured using a confocal microscope.

**Figure 5 ijms-21-04800-f005:**
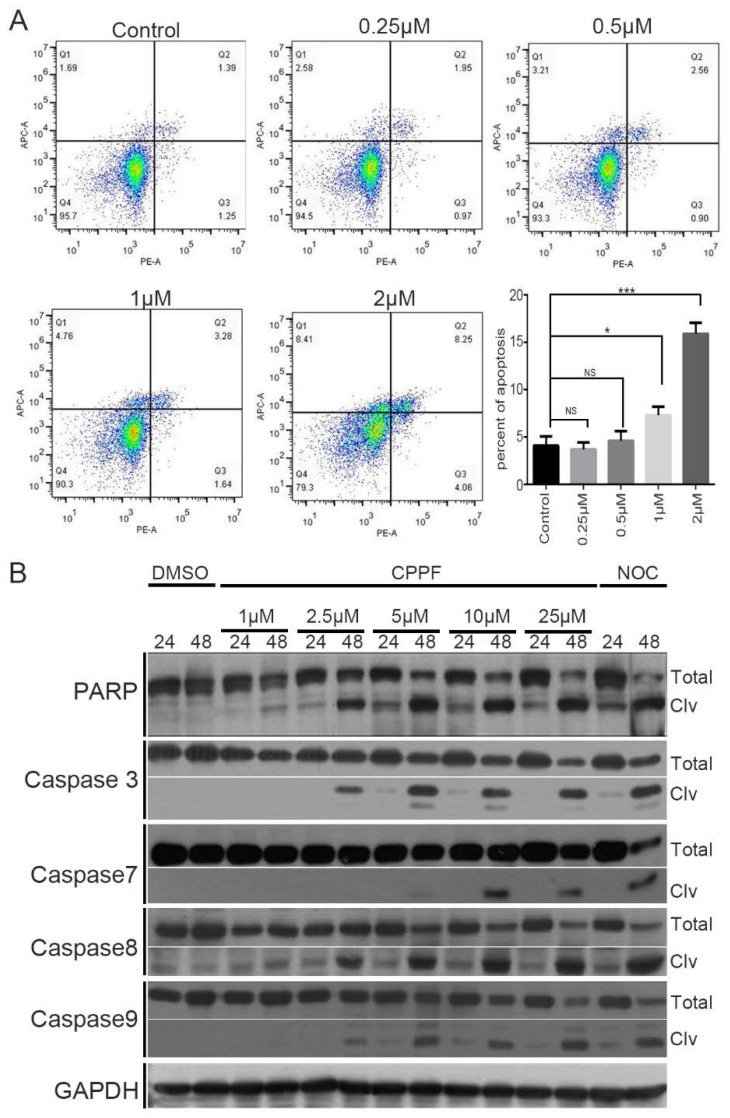
CPPF causes apoptotic cell death. (**A**) HeLa cells treated with various concentrations of CPPF for 48 h. Cells were harvested and stained with propidium iodide (PI) and Annexin V-APC. Apoptosis was analyzed by flow cytometry. Error bars represent means ± SD from three independent experiments, * *p*-value < 0.05 and *** *p*-value < 0.0005, NS: Not significant (**B**) Western blot data showed the accumulation of apoptosis marker proteins such as cleaved PARP, cleaved Caspases 3, 7, 8 and 9. These data are representative of three independent experiments. The concentration of nocodazole (NOC) was 100 ng/mL. GAPDH was used as a loading control.

**Figure 6 ijms-21-04800-f006:**
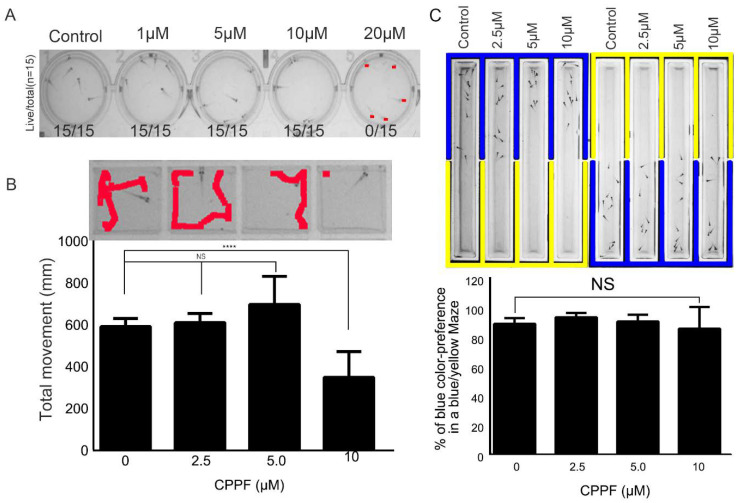
CPPF has low toxicity in zebrafish. (**A**) Five DPF zebrafish were incubated in a 12-well plate and treated with various concentrations of CPPF. After 24, 48 and 72 h, images were captured and analyzed to determine survival (*n* = 15); (**B**) Five DPF zebrafish were incubated in a 96-well plate and treated with various concentrations of CPPF. After 1 h adaptation, the zebrafish movements were recorded for 4 h (*n* = 6). The total distance traveled was measured by Ethovision software. Error bars represent means ± SD of 6 fish movement, **** *p*-value < 0.0001 and NS: Not significant; (**C**) Five DPF zebrafish were incubated in a blue-yellow chamber and treated with 0, 2.5, 5 or 10 µM CPPF. After 1 h in the dark, zebrafish were exposed to 123 lux of light for 1 h and record. At 10 min interval, the location and number of fish were logged. Error bars represent means ± SD of 20 fish movement, NS: Not significant.

**Figure 7 ijms-21-04800-f007:**
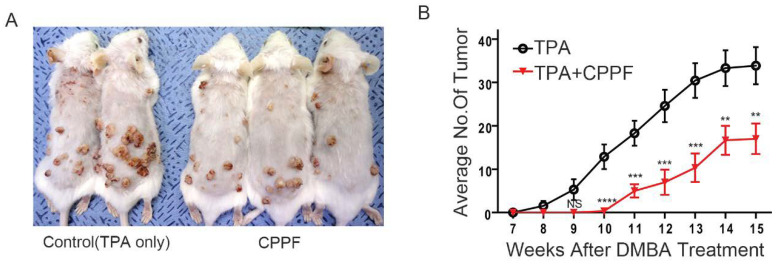
CPPF shows antiproliferation and prevention effect in vivo model. (**A**) The cancer prevention effect of CPPF in two-step skin cancer model. FVBN mice were topically treated with DMBA once and treated with TPA only or TPA plus 0.5 µM of CPPF 3 times per week. Images were taken at the end of the experiment; (**B**) The average number of skin papilloma was measured weekly. The data are presented as mean± SD, ** *p*-value < 0.005, *** *p*-value < 0.001, **** *p*-value < 0.0001. NS: Not significant.

**Table 1 ijms-21-04800-t001:** IC_50_ of multidrug-resistant cell lines.

Cell Lines	CPPF (µM)	Paclitaxel (µM)	Colchicine (µM)
K562	0.386 ± 0.067	0.008 ± 0.01	0.0123 ± 0.002
K562/ADR	0.326 ± 0.047 (0.84)	1.229 ± 0.243 (153.6)	0.3813 ± 0.101 (31)
MCF7	7.95 ± 0.714	0.004 ± 0.002	0.012 ± 0.008
MCF7/ADR	1.91 ± 0.056 (0.24)	1.573 ± 0.063 (393.2)	0.446 ± 0.066 (37.1)

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
