# Peer review of "CPPF, A Novel Microtubule Targeting Anticancer Agent, Inhibits the Growth of a Wide Variety of Cancers"

_ijms, 2020, doi:10.3390/ijms21134800_

Round 1
Reviewer 1 Report
Majors
1. In section 2.5, CPPF seemed to be a weak apoptosis inducer. The authors may consider discussing this and propose a potential future direction.
2. CPPF appears to show collateral selectivity in Table 1, the authors may consider to discuss this and provide a hypothetical mechanism.
Minors,
1, Line 46, " serve critical functions", "exert critical functions" is better.
2, Please indicate the origin of this compounds library.
3, I noticed that the structure of CPPF in SI is shown as HCl salt, while there is no H signal for HCl?
4, Line 224, the title of Table 1 is incorrect. Please include RF in this table. And have you tested the combination of CPPF with either Pac or Col?
5, The authors may consider adding Fig. S1 and S2 in the main text.
6, Please revise the discussion accordingly.
Author Response
Reviewer #1
Majors
- In section 2.5, CPPF seemed to be a weak apoptosis inducer. The authors may consider discussing this and propose a potential future direction.
Thank you for your comment. We have adjusted the conclusion section (line 445-448) to acknowledge this fact and to speak about our plans to not only improve the ability of CPPF to induce apoptosis but also to reduce its IC50 and improve drug delivery by developing derivates of CPPF.
- CPPF appears to show collateral selectivity in Table 1, the authors may consider to discuss this and provide a hypothetical mechanism.
Thank you for your comment. The discussion section of the manuscript (line 303-309) has been modified to speculate on a possible mechanism used by CPPF to overcome drug resistance. In short, there is a likelihood that this mechanism is independent of the ability of CPPF to bind tubulin. Perhaps, CPPF binds to and inhibits an MDR related protein or a drug efficacy protein. Having a tubulin-independent mechanism could explain how CPPF shares a cancer inhibitory phenotype with other MTAs but can overcome their drug resistance.
Minors,
- Line 46, "serve critical functions", "exert critical functions" is better.
Thank you for your comment. We have made this change in the manuscript.
- Please indicate the origin of this compounds library.
Thank you for your comment. The compound library used for this study came from the Korea Chemical Bank. The manuscript has been changed to state this fact.
- I noticed that the structure of CPPF in SI is shown as HCl salt, while there is no H signal for HCl?
Thank you for your comment. As you can see from the Mass spectrum results of CPPF HCl, the molecular weights were measured to be 333.3 and 335.2. These values were due to the isotope value of Cl (standard atomic weight; 35.45, isotopes, 35Cl: 37Cl =3:1) these mean that CPPF HCl salt form was detected. It can be seen that even though the H signal was not confirmed by measuring Cl NMR, in nature, Cl is not present alone, thus, H is included in the molecular weight measurement value of the synthetic CPPF HCl.
- Line 224, the title of Table 1 is incorrect. Please include RF in this table. And have you tested the combination of CPPF with either Pac or Col?
Thank you for your comment. You are correct that the title of table 1 had a typo. This has been corrected to read “Table. IC50 of multidrug-resistant cell lines”. As requested, the resistant factors (RF) have been added to the table. We have not tested a CPPF and Pac or Col combination, but it is a good suggestion that we will pursue in the future.
- The authors may consider adding Fig. S1 and S2 in the main text.
Thank you for your comment. After further review, we followed your suggestion and moved the zebrafish data from supplemental Figure 1 to Figure 6. However, we compared in vivo experiments and found that the anticancer activity of CPPF in the skin cancer model was greater than the xenograft model, so we choose to leave the xenograft data in supplemental Figure 2.
- Please revise the discussion accordingly.
Thank you for all your comments and suggestions. The discussion section of the manuscript .(line 280, line 302, line 303~308, line 309 line 310~311) has been modified to address your comments and concerns.
***
All line were refer to origin file (before revised)
***
Revised can be observed following memo of new manuscripts.
Reviewer 2 Report
The authors demonstrated that CPPF, a novel microtubule targeting anticancer agent, inhibited the growth of a wide variety of cancers. Several in vitro and in vivo models were applied to testify the effect and mechanism of CPPF. This is a well designed and well-performed study.
Minor concerns:
- The table caption of table 1 should be corrected to "IC50 of CPPF in the cancer cell lines".
- The manuscript should be polished by a native English speaker.
Author Response
Reviewer #2
Comments and Suggestions for Authors
The authors demonstrated that CPPF, a novel microtubule targeting anticancer agent, inhibited the growth of a wide variety of cancers. Several in vitro and in vivo models were applied to testify the effect and mechanism of CPPF. This is a well designed and well-performed study.
Minor concerns:
- The table caption of table 1 should be corrected to "IC50 of CPPF in the cancer cell lines".
Thank you for your exact point. Because the title of the table was not correct, 50 were changed to IC50. As well as, we have included the resistance factor (RF) in the table.
- The manuscript should be polished by a native English speaker.
Although, we already confirmed all the manuscript by native speaker once. If you thought our manuscript were not enough English gramma. We are going to get the MDPI English service.
***
Revised can be observed following memo of new manuscripts.
Reviewer 3 Report
Despite the extensive work in this manuscript. However, it contains various concerns that affect the validity of the study. In addition, it needs extensive organizations and revisions as follows:
Major concerns:
1. The statistical analysis is necessary to validate the findings of the study but it is missed here. Thus, a separate section clarifying the model used for statistical analysis, the post hoc test used for significance, and the types of software used should be added. Additionally, statistical analysis should be clearly described in the results section.
2. The results of the toxicity study in zebrafish have been described in the results section. However, no information about this experiment has been mentioned in the material and methods section. Also, in the methods section of Carcinogen Induced Skin Cancer, the authors mentioned that "skin tumors were removed for histological and biochemical studies". Where are the results of these histological and biochemical studies?
3. The results section is a major concern in the current study. The results in its present are a part of the discussion and a part of the material and methods. Hence, the authors should first transfer all paragraphs with references (e.g. lines 105-114) to the discussion section. Also, all paragraphs describing the methods used (lines 133-135) to the material and methods section. Then, the results should be rewritten after proper statistical analysis.
4. The discussion in its present form is a serious drawback of the study. The results in its present are a part of the results and a part of material and methods. Consequently, it is highly recommended to rewrite the discussion after fortification with the paragraph transferred from the result section.
5. There is a problem in using the abbreviations throughout the manuscript. The full term must be introduced upon the first mentioning followed by its abbreviation in parentheses: From then on, the abbreviation must be used exclusively and throughout. E.g. in line 43: microtubule targeting agents (MTAs) then the full term has been repeated again in line 51.
Specific comments:
1. Keywords: CPPF should be added.
2. Introduction: the authors give very little information on 5-5 (3-Chlorophenyl)-N-(3-pyridinyl)-2-furamide (CPPF). On what hypothesis the authors suggested its probable anticancer activity.
3. Material and methods: much important information is missed like:
- Line 364: what are the concentrations used?
- Carcinogen induced skin cancer: what is the total number of mice used, male or female, and the experimental groups.
- Line 441: what are the measured biochemical indicators?
Author Response
Reviewer #3
Despite the extensive work in this manuscript. However, it contains various concerns that affect the validity of the study. In addition, it needs extensive organizations and revisions as follows:
Major concerns:
1. The statistical analysis is necessary to validate the findings of the study but it is missed here. Thus, a separate section clarifying the model used for statistical analysis, the post hoc test used for significance, and the types of software used should be added. Additionally, statistical analysis should be clearly described in the results section.
Thank you for your comments. As you have suggested, we have added a statistical analysis section to the Materials and Methods.( section 4.15) In this section, we clearly state the statistical analysis were a nonparametric t-test, which has done using the Graphpad Prism 6.0 program. In addition, we added statistical analysis for each graph (Fig 1C, Fig 2C, Fig 3B, Fig 5A, Fig 6B,C, Fig7 B). The statistical analysis is now described in each figure legend.
The results of the toxicity study in zebrafish have been described in the results section. However, no information about this experiment has been mentioned in the material and methods section. Also, in the methods section of Carcinogen Induced Skin Cancer, the authors mentioned that "skin tumors were removed for histological and biochemical studies". Where are the results of these histological and biochemical studies?
In the original manuscript, the zebrafish data was described in the supplemental section. For clear explanation we have moved this data to the main text and data from supplemental Figure 1 has been moved to Figure 6. In addition, methods describing the zebrafish experiments have been added to the Materials and Methods (section 4.12, 4.13). We had a writing mistake in “skin tumors were removed for histological and biochemical studies". We only analyzed these experiment data by counting of skin cancer number. So we exchanged the sentence into “sacrificed”.
The results section is a major concern in the current study. The results in its present are a part of the discussion and a part of the material and methods. Hence, the authors should first transfer all paragraphs with references (e.g. lines 105-114) to the discussion section. Also, all paragraphs describing the methods used (lines 133-135) to the material and methods section. Then, the results should be rewritten after proper statistical analysis.
Thank you for your scientific comments, we deleted lines 105-114 and 133-135 parts and then rewrote in discussion parts. As considering of your comments, we also changed result parts (line 195, line 230, line 239, line 247, line 257).
The discussion in its present form is a serious drawback of the study. The results in its present are a part of the results and a part of material and methods. Consequently, it is highly recommended to rewrite the discussion after fortification with the paragraph transferred from the result section.
Thank you for your comment. As discussed in the previous comment, we have made the changes to the discussion section after incorporating the specified paragraphs from the result section (line 280, line 300, line 302, line 303~308, line 309, line 310~311).
There is a problem in using the abbreviations throughout the manuscript. The full term must be introduced upon the first mentioning followed by its abbreviation in parentheses: From then on, the abbreviation must be used exclusively and throughout. E.g. in line 43: microtubule targeting agents (MTAs) then the full term has been repeated again in line 51.
Thank you for your comment, we changed all of wrong abbreviation include your points such as line43, and line 51. (In addition, line70, line77, line 78, line 83, line 162, line193, line 215, line 216, line 225, and line 290).
Specific comments:
1. Keywords: CPPF should be added.
Thank you for to your comment. We added CPPF to the keywords (Line 34).
- Introduction: the authors give very little information on 5-5 (3-Chlorophenyl)-N-(3-pyridinyl)-2-furamide (CPPF). On what hypothesis the authors suggested its probable anticancer activity.
Thank you for your comment. In the introduction(line 74)., we newly described that we were screening a small molecule library from the Korea Chemical Bank looking for potential anticancer drugs and identified CPPF for its ability to inhibit the growth of cancer cells.
- Material and methods: much important information is missed like:
- Line 364: what are the concentrations used?
Thank you for your comment. We have modified Materials and Methods 4.4 (line 364), 0, 0.1µM, 0.3 µM, 1 µM, 5 µM, 10 µM were treated for MTT assay
-Carcinogen induced skin cancer: what is the total number of mice used, male or female, and the experimental groups.
Thank you for your comment. We described in Materials and Methods 4.14 (line 434) that male mice were used and in each group, more than 20 mice were used.
- Line 441: what are the measured biochemical indicators?
As we described in Materials and Methods, the papillomas were measured the size. Over 2mm of diameter of papilloma was considered as positive for this assay.
***
All line were refer to origin file (before revised)
***
Revised can be observed following memo of new manuscripts.
Round 2
Reviewer 1 Report
Thank you for the revision. It's my opinion that it can be published in the current form.
Author Response
Thank you for the Review of our manuscripts.
For your Review, substantially improved the quality and presentation of our manuscript
Reviewer 3 Report
Despite the significant improvements in the manuscript, but some concerns need to b addressed to fit for publication as follows:
1. The discussion stills a major concern in the study. The authors only transferred the paragraphs from the results as previously directed without revising the whole discussion as also directed. Consequently, the authors are highly recommended to discuss the estimated parameters with the probable interpretations correlated to the proposed biological activity of CPPF.
2. Line 24: replace "CPPF" with "5-5 (3-Chlorophenyl)-N-(3-pyridinyl)-2-furamide (CPPF)".
3. Line 447: what do" more than 20 mice" mean? The authors should mention the exact number of mice used in the study.
Author Response
Despite the significant improvements in the manuscript, but some concerns need to b addressed to fit for publication as follows:
- The discussion stills a major concern in the study. The authors only transferred the paragraphs from the results as previously directed without revising the whole discussion as also directed. Consequently, the authors are highly recommended to discuss the estimated parameters with the probable interpretations correlated to the proposed biological activity of CPPF.
à Thank you for your comments, we tried to appeal about drugable possiblility the compound we studied at the discussion part. As well as, even if it has not been verified, it was introduced in future plans to explain the possibility of the substance. So the line 272, 276,300, 305 were rewrote and added.
- Line 24: replace "CPPF" with "5-5 (3-Chlorophenyl)-N-(3-pyridinyl)-2-furamide (CPPF)".
à Thank you for your comments, line 24 CPPF were changed 5-5 (3-Chlorophenyl)-N-(3-pyridinyl)-2-furamide (CPPF).
- Line 447: what do" more than 20 mice" mean? The authors should mention the exact number of mice used in the study.
à Thank you for comments, we analyzed data form 20 mice for each group, so exact number was 20 mice and the line 447 ‘more than’ were deleted.